# Liquid Chromatography ICP-MS to Assess the Stability of ^175^Lu- and ^nat^Ga-Based Tumor-Targeting Agents towards the Development of ^177^Lu- and ^68^Ga-Labeled Radiopharmaceuticals

**DOI:** 10.3390/pharmaceutics16030299

**Published:** 2024-02-21

**Authors:** Rahel H. Wallimann, Heloïse Hensinger, Cristina Müller, Roger Schibli, Rainer Kneuer, Patrick Schindler

**Affiliations:** 1Biomedical Research, Novartis, 4056 Basel, Switzerland; rahel.wallimann@novartis.com (R.H.W.); heloise.hensinger@novartis.com (H.H.); rainer.kneuer@novartis.com (R.K.); 2Department of Chemistry and Applied Biosciences, ETH Zurich, 8093 Zurich, Switzerland; cristina.mueller@psi.ch (C.M.); roger.schibli@psi.ch (R.S.); 3Center for Radiopharmaceutical Sciences, Paul Scherrer Institute, 5232 Villigen, Switzerland

**Keywords:** SEC-ICP-MS, RPC-ICP-MS, PSMA, metal conjugates, nuclear theranostics

## Abstract

In recent years, nuclear medicine has gained great interest, partly due to the success story of [^177^Lu]Lu-PSMA-617 (Pluvicto^TM^). Still, in-depth preclinical characterization of radiopharmaceuticals mainly happens at centers that allow working with radioactive material. To support the development of novel radiopharmaceuticals, alternative non-radioactive characterization assays are highly desirable. The aim of this study was to demonstrate that inductively coupled plasma mass spectrometry (ICP-MS) associated with a chromatographic system can serve as a surrogate for the classical high-performance liquid chromatography (HPLC)-radiodetector combination for preclinical in vitro characterization of non-radioactive metal-labeled analogs of radiopharmaceuticals. In this proof-of-concept study, we demonstrate the applicability of HPLC–ICP-MS by assessing the stability of ^175^Lu- and ^nat^Ga-labeled prostate-specific membrane antigen (PSMA)-targeting peptidomimetics, single domain antibody (sdAb) conjugates, and monoclonal antibody (mAb) conjugates. ^175^Lu-labeled DOTAGA-conjugated and ^nat^Ga-labeled NODAGA-conjugated sdAbs and mAbs showed the highest stability with >90% still intact after 24 h. The peptidomime-tics [^175^Lu]Lu-PSMA-617 and [^nat^Ga]Ga-PSMA-11 showed identical in vitro serum stability as it was reported for their corresponding radioligands with >99% intact species after 24 h incubation in mouse serum, demonstrating the reliability of the method. Hence, the established HPLC–ICP-MS methods can support the development of novel radiopharmaceuticals in a classical pharmaceutical setting.

## 1. Introduction

Different radioactive isotopes are exploited in nuclear oncology, allowing for diagnostic imaging (e.g., gallium-68) and endoradiotherapy (e.g., lutetium-177) of malignant neoplasms. Combining the radionuclide with a targeting agent capable of selectively binding to specific antigens (over)expressed on cancer cells is fundamental, ensuring the radioactive material accumulates only in tumor lesions and not in healthy tissue. For that purpose, small molecules, peptides, and biologics must be functionalized with a bifunctional chelator (e.g., DOTAGA, NODAGA) that allows complexation of the radiometals and linker attachment using synthetic (e.g., maleimide chemistry) or enzymatic (e.g., sortase catalyzed ligation) conjugation chemistries [1,2,3,4,5,6,7]. To avoid unspecific radionuclide accumulation in healthy tissue, the radiometallated conjugate must be stable and not degrade while circulating in the bloodstream. Stability is typically assessed using various in vitro assays in conjunction with high-performance liquid chromatography (HPLC)-radiodetector combinations [8], thus incurring the regulatory and fiscal burdens associated with the use of radionuclides during the entirety of ligand development. The possibility of characterizing non-radioactive analog metal conjugates in classical research laboratories would bring a competitive advantage by facilitating the performance of these experiments. Non-radioactive alternative characterization techniques for the analysis of in vitro cell binding assays and even in vivo biodistribution studies have already been implemented [9,10,11]. One of the techniques described in the literature is inductively coupled plasma mass spectrometry (ICP-MS), which is known for its extremely high sensitivity (in the ng L^−1^ range) and its wide dynamic range (up to 12 orders of magnitude) [12]. Compared to other approaches using mass spectrometry, ICP-MS is the only method that solely detects the metal payload and, therefore, represents the situation as it would be when detecting the emitted γ-radiation of a radionuclide [13]. The goal of this study was to demonstrate that ICP-MS combined with reversed-phase chromatography (RPC) and size exclusion chromatography (SEC) can be used as a surrogate method for radio-HPLC to investigate the in vitro stability of selected metal conjugates.

In this work, we evaluated prostate-specific membrane antigen (PSMA)-targeting agents as a commercially successful example of targeted radionuclide therapy. Specifically, the peptidomimetics PSMA-11 (ABX advanced biomedical compounds GmbH, Radeberg, Germany) and PSMA-617 (Pluvicto^TM^, MedChemExpress, Monmouth Junction, NJ, USA), labeled with stable isotopes of gallium and lutetium, respectively, were chosen as reference compounds [14,15] since they have been extensively characterized previously by radiometric methods. Additionally, in vitro serum stability assays with biologic metal conjugates were performed, as these commonly suffer from reduced stability as compared to small molecules depending on the selected conjugation chemistry or the chelator used [4,5,6,16]. The single domain antibody (sdAb) JVZ-007 [17] and the humanized monoclonal antibody (mAb) J591 [18] were conjugated with a chelator by maleimide chemistry, which is widely used in the development of antibody–drug conjugates. However, the formed thiosuccinimide linkage—a Michael acceptor—is hydrolytically unstable and prone to react with endogenous thiols present in biomolecules such as serum albumin [4,6,16]. In contrast, the sdAb and mAb were conjugated either by sortase-catalyzed ligation or by random lysine conjugation, resulting in a stable amide and a stable thiourea bond, respectively.

By using the same methods, the chemical stability of metal–chelator complexes may be assessed in a non-radioactive form, addressing another highly relevant question regarding the development of novel radiopharmaceuticals. As a proof of principle, we examined DOTAGA- and NODAGA-chelators, as they are well known in the field of nuclear oncology, and they were already used in clinics in combination with PSMA-targeting agents [19,20,21]. The DOTAGA-chelator is known to form stable complexes with lutetium-175, but it is not ideal for complexation with gallium ions. On the other hand, the NODAGA chelator forms gallium complexes with high stability; however, it cannot be used to complex lutetium-175 (Figure 1) [22].

## 2. Materials and Methods

### 2.1. Preparation of Metal-Labeled Peptidomimetics

Labeling of PSMA-617 (MedChemExpress, Monmouth Junction, NJ, USA) with the stable isotope lutetium-175 was performed as previously described [11]. In brief, PSMA-617 dissolved in 50 mM sodium acetate buffer (NaOAc, Sigma Aldrich Chemie, Steinheim, Germany), pH 5.5, was labeled with [^175^Lu]LuCl_3_ (Sigma Aldrich Chemie, Steinheim, Germany) in 0.05 M hydrochloric acid (HCl, VWR International S.A.S., Fontenay-sous-Bois, France) by adding one equivalent of the metal salt to the ligand solution. The reaction mixture was incubated at 95 °C for 10 min to obtain quantitative metal labeling. [^nat^Ga]Ga-PSMA-11 (ABX advanced biochemical compounds, Radeberg, Germany) was commercially available as a powder and dissolved in 50 mM NaOAc buffer, pH 3.7, at a concentration of 1 mM. The identity of the products was confirmed by liquid chromatography–mass spectrometry (LC–MS). The metal-labeled peptidomimetics were used for the establishment of the coupling of RPC to ICP-MS and for in vitro serum stability experiments.

### 2.2. Conjugation and Metal-Labeling of sdAbs and mAbs

The sequence of the anti-PSMA-sdAb was described by Chatalic et al. [17]. The sdAb was derivatized with a C-terminal cysteine and a sortase-his tag, respectively, to allow site-specific conjugation with a bifunctional chelator. The anti-PSMA-sdAb modified with a C-terminal cysteine was conjugated with the commercially available chelator 2,2′,2″-(10-(1-carboxy-4-((2-(2,5-dioxo-2,5-dihydro-1H-pyrrol-1-yl)ethyl)amino)-4-oxobutyl)-1,4,7,10-tetraazacyclododecane-1,4,7-triyl)triacetic acid (maleimide-DOTAGA, CheMatech, Dijon, France) and 2,2′-(7-(1-carboxy-4-((2-(2,5-dioxo-2,5-dihydro-1H-pyrrol-1-yl)ethyl)amino)-4-oxobutyl)-1,4,7-triazonane-1,4-diyl)diacetic acid (maleimide-NODAGA, abcr GmBH, Karlsruhe, Germany), respectively, following the procedure for maleimide conjugation as previously described by Chatalic et al. [17]. The derivate with the sortase recognition sequence LPETGG-His6 was conjugated with GGGK-DOTAGA, which was prepared in-house from commercially available precursors. Conjugation via sortase-catalyzed ligation was performed as previously described by Massa et al. [23]. The DOTAGA-conjugated anti-PSMA-sdAbs were quantitatively and stoichiometrically labeled to obtain the final [^175^Lu]Lu-DOTAGA-cys-mal-anti-PSMA-sdAb-GGC and [^175^Lu]Lu-DOTAGA-LPETGG-anti-PSMA-sdAb, respectively, by adding an excess of lutetium-175 as [^175^Lu]LuCl_3_ in 0.05 M HCl and allowing the reaction mixture to be incubated at 37 °C for 3 h. Metal labeling was followed by purification using a PD-10 G-25 column (Cytiva, Buckinghamshire, UK) and 50 mM NaOAc buffer, pH 5.5, as eluent. Labeling of the DOTAGA-cys-mal- and the NODAGA-cys-mal-conjugated anti-PSMA-sdAb with the stable isotopes of gallium required a lower pH than ^175^Lu-labeling. Therefore, the buffer of the conjugates was exchanged with chelexed (Bio-Rad, Hercules, CA, USA) 50 mM NaOAc buffer, pH 3.7, using Amicon^®^ centrifugal filter units (Merck KGaA, Darmstadt, Germany). Quantitative ^nat^Ga-labeling was performed by the addition of an excess of the stable isotopes of gallium as [^nat^Ga]GaCl_3_ (ThermoFisher, Kandel, Germany) in 0.05 M HCl, and the reaction mixture was shaken at 12 °C overnight, followed by purification using a PD-10 G25 column and phosphate-buffered saline (PBS, Biowest, Bradenton, FL, USA), pH 7.4, as the mobile phase. The final products were characterized by analytical SEC and LC–MS (Appendix A).

The humanized anti-PSMA-mAb J591 [18,24] was modified with engineered cysteines (for a drug antibody ratio of four, mutations mVh_E12C_S375C) to allow site-specific conjugation with the commercially available chelators maleimide-DOTAGA or maleimide-NODAGA, respectively. Conjugation via maleimide chemistry was performed as previously described by Coumans et al. [25]. Random conjugation of the anti-PSMA-mAb via lysine was performed following the procedure described by Le Bihan et al. [26] using the commercially available chelators *p*-NCS-bz-DOTAGA (CheMatech, Dijon, France) and *p*-NCS-bz-NODAGA (CheMatech, Dijon, France), respectively. Quantitative metal labeling was achieved by incubating the conjugates with an excess of the stable isotopes of lutetium and gallium, respectively, as described above. Metal labeling was followed by purification using a Superdex 200 Increase 10/300 GL column (Merck KGaA, Darmstadt, Germany), PBS, pH 7.4, as the mobile phase, and an isocratic flow of 0.3 mL/min. The final metal conjugates were concentrated using Amicon^®^ centrifugal filter units. Identification of the final products was made by LC–MS and analytical SEC. The degree of labeling for the metal conjugates was determined by direct infusion of ICP-MS (Appendix A). The synthesized metal conjugates were used as a reference for establishing the SEC-ICP-MS coupling procedure and for in vitro serum stability experiments.

Commercially available bovine serum albumin (BSA, Sigma-Aldrich, Steinheim, Germany) was conjugated with the commercially available [^165^Ho]Ho-*p*-SCN-bn-DOTA (Macrocyclics, Plano, TX, USA) by random conjugation via lysine [26]. The identity of the ^165^Ho-labeled BSA was confirmed by analytical SEC and LC–MS (Appendix A). The final product was used as an internal standard for the in vitro serum stability assays.

### 2.3. Instrumentation

All ICP-MS measurements were performed on a triple quadrupole ICP-MS (iCAP TQ, Thermo Fisher, Reinach, Switzerland). The ICP-MS was equipped with a perfluoroalkoxy liquid chromatography integrated capillary valve nebulizer (Elemental Scientific, Inc., Ohama, NE, USA) and a cyclonic quartz spray chamber (Thermo Fisher Scientific, Reinach, Switzerland), with the spray chamber temperature set at 2.7 °C. Before the measurements, the ICP-MS was thermally equilibrated by continuously running a 2% (*v/v*) nitric acid (HNO_3_, purified by redistillation, ≥99.999% trace metal basis, Merck KGaA, Darmstadt, Germany) solution for 30 min. Parameters, including nebulizer gas flow, lens voltage, and ICP radio frequency power, were optimized by autotuning the system to achieve adequate sensitivity. Autotuning of the instrument was carried out per manufacturer’s instructions using the iCAP Q/TQ TUNE solution (containing barium, bismuth, cerium, cobalt, holmium, indium, lithium, magnesium, titanium, uranium, and yttrium, all at concentrations of 1.00 ± 0.01 μg/L, in 2% (*v/v*) ultra-pure HNO_3_/0.05% (*v/v*) ultra-pure HCl, Thermo Fisher Scientific, Ward Hill, MA, USA). Typical operating conditions for the ICP-MS are summarized in Table 1. The detection of the analyte ions was performed in single quadrupole (SQ) mode. The analysis of the ^175^Lu-labeled ligands was performed using helium as a non-reactive gas (SQ-He) for focusing and detection of the lutetium-175 ions, whereas for the measurements of the ^nat^Ga-labeled ligands, the kinetic energy discrimination (KED) mode was applied to reduce spectral interferences. The latter was analyzed only for the isotope gallium-71, which suffers from fewer spectral interferences compared to the isotope gallium-69. RPC and SEC were performed using a Vanquish flex ultra-performance liquid chromatography system (Thermo Fisher Scientific, Reinach, Switzerland) equipped with a quaternary pump, a split sampler, and a variable wavelength detector (Vanquish flex, Thermo Fisher Scientific, Reinach, Switzerland). When the ICP-MS was used in combination with RPC, oxygen (5%) was added to the argon gas flow to oxidize the organic matter present in the mobile phase [27] and to pyrolyze potential carbon deposits on the cones [28]. The addition of oxygen to the argon gas flow and the presence of the organic solvent load required the use of more robust platinum samplers and skimmer cones [29,30]. Thullium-169 (1 µg/L in 2% (*v/v*) ultra-pure HNO_3_, 0.05 mL/min flow) was constantly added as a reference element via a T-piece after the column by a separate isocratic pump to monitor the performance of the system and was measured in SQ-He mode.

### 2.4. Analytical Method Development

For RPC-ICP-MS analysis, the system was equipped with a C-18 reversed-phase column (Halo Peptide, 160 Å, ES-C18, 2.7 µm, 1.0 mm I.D. × 75 mm length, Waters, Wilmington, DE, USA). The mobile phase consisted of 0.1% trifluoroacetic acid (TFA, (*v/v*), Sigma Aldrich Chemie, Steinheim, Germany) in methanol (MeOH, solvent A, Biosolve Chimie SARL, Dieuze, France) and 0.1% TFA (*v/v*) in milli-Q water (H_2_O, solvent B) with a linear gradient of 2–80% solvent A over 10 min. A low flow rate of 0.1 mL/min was applied to avoid the formation of carbon crystals on the ICP-MS cones while maintaining peak separation. The effect of MeOH on the ionization rate of the ICP-MS was monitored by the thulium-169 solution that was added as a direct reference to the ICP-MS system. When diluting product samples in 0.1% TFA (*v/v*) in a mixture of H_2_O:MeOH (1:49, *v/v*), the adhesion of the product samples to the HPLC vials and the tubing of the system resulted in insufficient recovery. This was avoided by diluting [^175^Lu]Lu-PSMA-617 and [^nat^Ga]Ga-PSMA-11 in rat blood extract. Rat blood extract was prepared by vigorous mixing of rat blood (Charles River Laboratories, Sulzfeld, Germany) with a five-fold excess of a mixture of acetonitrile (ACN, Sigma Aldrich Chemie, Steinheim, Germany) and H_2_O (1:1, *v/v*), followed by centrifugation (5 min; 20′800 rcf; 4 °C) and dilution of the supernatant with ACN, MeOH, and 0.1% TFA in H_2_O (1:1:1:22, *v/v/v/v*). Diethylenetriaminepentaacetic acid (DTPA, 254 µM, Sigma Aldrich Chemie, Steinheim, Germany) or ethylenediaminetetraacetic acid (EDTA, 17 µM, Sigma Aldrich Chemie, Steinheim, Germany) was added to the sample preparation to complex uncoordinated metal ions. This prevented any uncoordinated metal ions from interacting with the stationary phase of the column. Mixtures of [^175^Lu]Lu-PSMA-617 and lutetium-175 ions and of [^nat^Ga]Ga-PSMA-11 and gallium-69/gallium-71 ions, respectively, were analyzed by the established methods for RPC-ICP-MS to demonstrate separation of the metal conjugates from uncoordinated metal ions. Calibration curves for [^175^Lu]Lu-PSMA-617, [^nat^Ga]Ga-PSMA-11, and the metal ions complexed by either DTPA or EDTA were analyzed by RPC-ICP-MS to demonstrate the robustness of the developed chromatographic conditions. The calibration curves were established under the same conditions as the analysis of the in vitro serum stability samples. They were used to define the linear range to allow relative quantification of the fractions obtained by the in vitro serum stability assays. The standards for the calibration curves were prepared by serial dilution of stock solutions in mouse serum extract to obtain final concentrations of the ligands between 2 nM and 5 µM and for the metal ions between 0.05 nM and 1.1 µM. To prepare the calibration curves of the metal ions, standard solutions of lutetium and gallium for ICP-MS (1 g/L solution in 2% (*v/v*) nitric acid, Merck KGaA, Darmstadt, Germany) were used. The injected volumes of the standards were 5 µL. For each metal-labeled peptidomimetic and metal ion, three independent calibration curves were prepared and analyzed by RPC-ICP-MS. The chromatograms were evaluated using Qtegra^TM^ software (version 2.10.4345.64, Thermo Fisher Scientific Inc., Reinach, Switzerland). The integrals of the individual peaks were recorded and plotted against the injected amount of metal expressed as pmol. To obtain the calibration curves, a simple linear regression analysis was applied using GraphPad Prism software (version 9). Before the analysis of the in vitro serum stability samples, a blank of mouse serum extract containing either DTPA (254 µM) or EDTA (17 µM) was injected to determine possible spectral interferences and metal-containing contaminations on the column.

The analysis of the sdAb metal conjugates (~13 kDa) was performed by SEC-ICP-MS. The column and the running conditions for SEC were tested and optimized for each metal conjugate to avoid possible shearing effects. The chromatographic system was equipped with a BEH protein column (BEH SEC, 125 Å, 1.7 µm, 4.6 mm I.D. × 150 mm length, Waters, Wilmington, DE, USA). Ammonium acetate buffer (20 mM, pH 5.0, Sigma Aldrich Chemie, Steinheim, Germany) was used as the mobile phase, and an isocratic flow at a flow rate of 0.4 mL/min was applied. The analysis of the mAb metal conjugates (~145 kDa) was conducted using a Premier Protein SEC column (Premier Protein SEC, 250 Å, 1.7 µm, 4.6 mm I.D. × 150 mm length, Waters, Wilmington, DE, USA) to achieve sufficient resolution for the separation of the intact mAb metal conjugate from metal-containing fractions co-eluting with serum proteins. The mobile phase consisted of ammonium acetate buffer (200 mM, pH 7.0, solvent A) and H_2_O (solvent B). An isocratic flow at a flow rate of 0.3 mL/min with 50% solvent A was applied. An aliquot of the biologic metal conjugate was diluted with 0.1% TFA (*v/v*) in H_2_O containing apomyoglobin (0.05 mg/mL, Sigma Aldrich Chemie, Steinheim, Germany) and either DTPA (254 µM) or EDTA (17 µM). The apomyoglobin was added to avoid adhesion of the metal conjugates to the HPLC vials and the tubing of the system. To confirm that the elaborated chromatographic conditions allow the separation of the metal-labeled biologic conjugates from the relevant lower and higher molecular weight species, mixtures of the biologic metal conjugates with metal ions complexed by EDTA and metal–chelator fragments were prepared and analyzed by SEC-ICP-MS.

The metal conjugates and lutetium-175 and gallium-69/gallium-71 ions complexed by DTPA and EDTA, respectively, were used to generate calibration curves for relative quantification of the metal-labeled fractions obtained by in vitro serum stability assays. The standards for the calibration curves were prepared by serial dilution of stock solutions to achieve final concentrations of 0.03 nM to 3.2 µM for the biologic metal conjugates and 0.05 nM to 1.1 µM for the metal ions. The calibration curves were prepared in mouse serum that was diluted in 0.1% TFA (*v/v*) in H_2_O containing either DTPA (254 µM) or EDTA (17 µM) to complex unbound metal ions. The injected volumes of the standards were between 5 and 15 µL. For each metal-labeled biologic conjugate and metal ion, three independent calibration curves were prepared and analyzed by SEC-ICP-MS. The chromatograms were evaluated using Qtegra^TM^ software (version 2.10.4345.64, Thermo Fisher Scientific Inc., Reinach, Switzerland). The integrals of the individual peaks were recorded and plotted against the injected amount of metal expressed as pmol. A simple linear regression analysis was applied using GraphPad Prism software (version 9) to obtain the calibration curves. A blank of mouse serum diluted in H_2_O containing either DTPA (254 µM) or EDTA (17 µM) was injected to determine possible spectral interferences and metal-containing contaminations on the column.

### 2.5. In Vitro Serum Stability Experiments

All in vitro serum stability experiments were performed in mouse serum (sterile filtered, Biowest, Bradenton, FL, USA) at 37 °C. [^175^Lu]Lu-PSMA-617, [^nat^Ga]Ga-PSMA-11, and the sdAb metal conjugates were incubated at a molar concentration of 5 µM. The mAb metal conjugates were incubated at a molar concentration between 0.5 µM and 1.5 µM. Aliquots were taken immediately after adding the metal conjugates to the serum, as well as at the time points 1 h, 2 h, 4 h, 6 h, and 24 h. The aliquots were stored at −80 °C to stop the reaction. The samples of the peptidomimetics were mixed with a six-fold excess of a mixture of ACN:H_2_O 7:3 (*v/v*) containing [^165^Ho]Ho-PSMA-617 (0.5 µM for [^175^Lu]Lu-PSMA-617 and 10 nM for [^nat^Ga]Ga-PSMA-11) for extraction of the ligands from the serum to allow subsequent analysis by RPC-ICP-MS. [^165^Ho]Ho-PSMA-617 was used for quantifying the recovery of the ligands from the serum (Appendix A). Samples of the biologic metal conjugates were diluted in 0.1% TFA (*v/v*) in H_2_O containing EDTA (17 µM) to complex unbound metal ions and [^165^Ho]Ho-DOTA-BSA (9 nM) as reference metal conjugate to reach final concentrations of 100–500 nM suitable for analysis by SEC-ICP-MS.

## 3. Results

### 3.1. Synthesis of Metal Conjugates

Quantitative labeling of PSMA-617 with lutetium-175 was achieved. The high chemical purity (≥99%) of the metal complexes was assessed by RPC-ICP-MS, confirming the absence of uncomplexed [^175^Lu]Lu^3+^ ions in the product solution. The identity of [^175^Lu]Lu-PSMA-617 and the commercially available [^nat^Ga]Ga-PSMA-11 were confirmed by LC–MS (detected masses m/z = 1′214.0 [M+H]^+^ for [^175^Lu]Lu-PSMA-617 and m/z = 1′012.3 [M+H]^+^ for [^nat^Ga]Ga-PSMA-11).

All sdAb and mAb metal conjugates were synthesized by either site-specific maleimide chemistry, site-specific sortase-catalyzed ligation, or random lysine coupling in sufficient yields for subsequent SEC-ICP-MS method development and in vitro serum stability assays. The identity of the metal conjugates was confirmed by analytical SEC and LC–MS (Appendix A). Chemical purity of ≥90% after metal labeling was confirmed by SEC-ICP-MS.

### 3.2. Analytical Method Development

Various compositions of the mobile phase and flow rates for the chromatography and the necessary amount of oxygen as additional gas for the nebulizer gas flow of the ICP-MS were investigated with the aim of reducing the formation of carbon crystals and salt crystals at the cones and the torch system. The optimized final conditions for RPC using 0.1% TFA (*v/v*) in MeOH instead of ACN and increasing the addition of oxygen gas to 5% resulted in almost negligible formation of carbon deposits while maintaining the sensitivity of the ICP-MS.

Using ammonium acetate buffer (20 mM to 100 mM) as a volatile buffer for SEC prevented the excessive formation of salt crystals on the ICP-MS system. Thus, the quality of the signal obtained by ICP-MS was preserved while maintaining peak separation on the SEC.

Robust methods were established to evaluate the in vitro stability of metal conjugates in mouse serum by RPC- and SEC-ICP-MS. Separation of ^175^Lu- and ^nat^Ga-labeled conjugates from relevant metabolites, such as metal ions or metal–chelator fragments co-eluting with serum proteins and metal-containing linker–chelator fragments, was demonstrated by the injection of a mixture of the relevant metal-labeled species. The peak retention times of the injected standards were used to identify the metal-labeled species obtained by the in vitro serum stability assays. Separation of the small molecular weight fractions, in particular, lutetium-175 complexed by DTPA or EDTA and [^175^Lu]Lu-DOTAGA-mal, could not be expected as the molecular weights of these molecules were not within the specified separation range of the column (1 kDa to 80 kDa). Calibration curves were used to define the linear range for relative quantification of the fractions obtained by the in vitro serum stability assays (Appendix A). The optimal composition of the extraction solvent was elaborated to ensure >90% recovery of the peptidomimetics from the mouse serum (Appendix A). To evaluate the in vitro serum stability of biologic metal conjugates, dilution of the samples in 0.1% TFA (*v/v*) in H_2_O containing EDTA (17 µM) was suitable for subsequent analysis by SEC-ICP-MS. This ensured 100% recovery of the metal conjugates and the relevant metal-containing fractions.

### 3.3. In Vitro Serum Stability in Mouse Serum

The chemical stability of the ^175^Lu- and ^nat^Ga-labeled PSMA-peptidomimetics of >99% was confirmed by in vitro serum stability assays analyzed by RPC-ICP-MS as described above (Appendix A).

SEC-ICP-MS analysis verified that conjugation of the sdAb-conjugate via sortase-catalyzed ligation resulted in the formation of a highly stable sdAb metal conjugate. This was demonstrated by the fact that ≥99% of the lutetium-175 of the [^175^Lu]Lu-DOTAGA-LPETGG-anti-PSMA-sdAb eluted with an identical peak retention time after synthesis and after 24 h incubation in mouse serum. In the case of [^175^Lu]Lu-DOTAGA-cys-mal-anti-PSMA-sdAb-GGC, approximately 10% of the total lutetium-175 (t_R_ = 2.4 min) eluted at a retention time identical to [^165^Ho]Ho-DOTA-BSA (t_R_ = 2.4 min), due to potential maleimide linker transfer to serum proteins such as albumin (Figure 2).

For the ^nat^Ga-labeled sdAb conjugates, SEC-ICP-MS analysis allowed the investigation of the chemical stability of two different gallium-69/gallium-71 chelator complexes. [^nat^Ga]Ga-DOTAGA-cys-mal-anti-PSMA-sdAb-GGC showed significant degradation over 24 h, where 68% of the metal eluted as small molecular-weight species. These small molecular weight species could be assigned to uncoordinated gallium ions complexed by EDTA based on the observation that the measured retention time was comparable to the one obtained for the standard [^nat^Ga]Ga-EDTA. For the sdAb metal conjugate bearing the NODGAGA-chelator, the amount of gallium ions complexed by EDTA was negligible, confirming that the NODAGA-chelator formed a more stable complex with gallium ions compared to the DOTAGA-chelator. For both ^nat^Ga-labeled sdAb conjugates, a peak with a similar retention time (t_R_ = 2.4 min) to [^165^Ho]Ho-DOTA-BSA (t_R_ = 2.4 min) was observed, which can be explained by potential maleimide linker transfer to serum proteins (Figure 3).

To further demonstrate the applicability of SEC-ICP-MS methods for the characterization of metal-labeled analogs of radiopharmaceuticals, in vitro serum stability assays were performed for the mAb metal conjugates obtained via random lysine conjugation (stable amide bond) or via cysteine maleimide conjugation (reversible thioether bond). The most stable chelators for the corresponding metals were selected for conjugation to allow investigation of the linker stability. As expected, the ^175^Lu- and ^nat^Ga-labeled mAb conjugates containing the stable *p*-NCS-bz linker showed the highest stability, with ≥99% of the metal eluting at the same retention time as that of the intact metal conjugate standard after 24 h incubation in mouse serum. The two mAb metal conjugates with the cysteine–maleimide linker showed lower stability than the *p*-NCS-bz linker. For the ^175^Lu-labeled DOTAGA anti-PSMA-mAb, 92% of the lutetium-175 was eluting with the same retention time after synthesis and after 24 h incubation, whereas for the ^nat^Ga-labeled NODAGA anti-PSMA-mAb, this percentage was reduced to 88%. In addition to the intact antibody metal conjugates, both conjugates showed 5–9% of metal (t_R_ = 4.2 min) eluting at an identical retention time to that of the injected [^165^Ho]Ho-DOTA-BSA (t_R_ = 4.2 min), indicating potential maleimide-linker transfer to serum proteins (Figure 4). For the ^175^Lu-labeled mAb conjugate containing the *p*-NCS-bz linker, <1% of low molecular weight species (t_R_ = 6.1 min) was observed, which could be attributed to uncoordinated metal ions and linker fragments, respectively (Figure 4A).

## 4. Discussion

Coupling ICP-MS with a chromatographic system allows structural estimations of the investigated molecules based on retention times and, thus, can serve as a surrogate method for radio-HPLC [31,32]. The application of HPLC-ICP-MS in the context of imaging probe discovery was already described by Boros et al. [28], where their study focused on the metabolite profiling of ^nat^Ga- and ^115^In-labeled peptides. We herein demonstrated that HPLC-ICP-MS can also be used to investigate the in vitro chemical stability of metal conjugates. Other than the use of radio-HPLC systems, combining the ICP-MS with RPC and SEC, respectively, may be challenging as each individual technique presents certain difficulties. In our study, the carbon content of the mobile phase used for RPC led to the formation of carbon deposits on the cones of the ICP-MS system, impairing its sensitivity. Different strategies previously described in the literature were applied to avoid the formation of carbon deposits on the cones. Among those are the addition of oxygen gas to the nebulizer gas flow, the reduction in the carbon content of the mobile phase by replacing ACN with MeOH, and the minimization of the chromatographic system flow from 0.2 mL/min to 0.1 mL/min [28,33]. When performing SEC, commonly nonvolatile buffers with high salt concentrations are used. Phosphate or acetate buffers are most suitable for the analysis of biologics to achieve strong separating power, but the nonvolatile salts of the buffer will likely cause ion suppression and spectral interferences on the ICP-MS [31,34]. Further, salt crystals can be formed in the nebulizer and torch of the ICP-MS, causing blockage of the system, which leads to a consequent reduction in sensitivity or even a complete loss of signal [33,35]. Hence, a volatile ammonium acetate buffer (20 mM to 100 mM) was found to be more suitable compared to PBS. The robustness of the chromatographic setup was confirmed by the obtained calibration curves for the metal conjugates and the metal ions complexed by DTPA and EDTA, respectively.

The established RPC- and SEC-ICP-MS methods allowed the investigation of the in vitro chemical stability of ^175^Lu- and ^nat^Ga-labeled biomolecules. The stabilities measured for the non-radioactive [^175^Lu]Lu-PSMA-617 and [^nat^Ga]Ga-PSMA-11 were equal to the stabilities reported in the literature for the radioactive [^177^Lu]Lu-PSMA-617 and [^68^Ga]Ga-PSMA-11 [36,37], demonstrating the reliability of the developed method. Analysis of in vitro serum stability assays by SEC-ICP-MS required only a dilution of the sample after incubation in mouse serum. Because no purification procedure such as immunoprecipitation was necessary, the risk of losing information about any metal-containing metabolites that might not be quantitatively extracted could be minimized. In this study, sdAb and mAb metal conjugates comprising linker chemistries (cysteine maleimide vs. sortase vs. *p*-NCS-bz) and metal–chelator complexes (^175^Lu- and ^nat^Ga-labeled DOTAGA vs. NODAGA) with variable stability profiles were used [4,6,16,22,38,39,40,41]. For all conjugates that were synthesized via maleimide chemistry, metal ions co-eluting with serum proteins were identified by SEC-ICP-MS. This finding can be explained either by serum proteins that bind uncoordinated metal ions or the potentially clipped-off maleimide-linker (or a fraction of it) or by direct thiol exchange via a retro-Michael process. The retro-Michael process is a widely known phenomenon for antibody–drug conjugates, where conjugation of a payload with an antibody by maleimide chemistry results in a reversible thioether bond that readily reacts with endogenous thiols [4,6,16]. Lutetium-175 or gallium-71 ions co-eluting with serum proteins were not observed for the metal conjugates synthesized using either sortase-catalyzed ligation or random conjugation via lysine and the *p*-NCS-bz linker, evidencing higher chemical long-term stability of these metal conjugates in mouse serum. Further, our SEC-ICP-MS method confirmed the anticipated superiority of the NODAGA-chelator compared to the DOTAGA-chelator regarding the stability of the formed gallium–ion–chelator complex [22,42].

One limitation of using non-radioactive metal conjugates to investigate their in vitro stability may arise from the fact that the effect of in vitro radiolysis is ignored. Radiolysis refers to either the direct damage of the vector molecule by radioactive particles emitted by the radionuclide or indirect damage of the biomolecule due to interaction with radicals formed during the radiolysis of water **[43,44,45]**. Consequently, the effect of in vitro radiolysis should be investigated using the radiolabeled version of the finally selected conjugates and using radio-HPLC for analysis. Another practical constraint of the technique is that information about the structure of the tumor-targeting agent is lost during ICP-MS analysis, as ICP-MS solely detects the metal of interest. Structural information about the tumor-targeting agent could be obtained when other techniques, such as nano-LC–MS, are used in complement to LC-ICP-MS to fully exploit the capabilities of the existing MS methodologies.

## 5. Conclusions

Our study has demonstrated that RPC- and SEC-ICP-MS can substitute certain assays, commonly performed with radiolabeled ligands using radio-HPLC, in a conventional industrial or academic setting where the work with radionuclides is not possible for regulatory or fiscal reasons. The implementation of (RPC-/SEC-) ICP-MS technologies would allow a faster pre-screening of novel ligands, chelators, and linkers based on improved linker or metal–chelator complex stability. This would consequently benefit the selection of the most promising candidates for the final evaluation involving radioactivity.

## Figures and Tables

**Figure 1 pharmaceutics-16-00299-f001:**
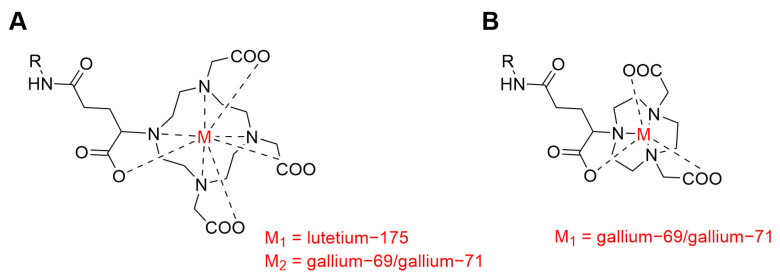
Chemical structures of metal complexes with the DOTAGA chelator and the NODAGA chelator. R stands for the selected tumor-targeting agent (anti-PSMA-sdAb or anti-PSMA-mAb). (**A**) metal-DOTAGA complex; (**B**) metal-NODAGA complex.

**Figure 2 pharmaceutics-16-00299-f002:**
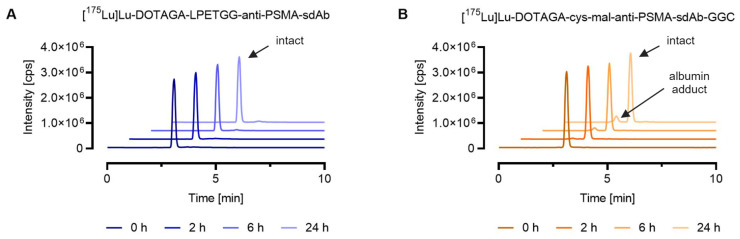
Comparison of the stability of ^175^Lu-labeled DOTAGA-anti-PSMA-sdAb conjugates synthesized via two different conjugation chemistries, sortase catalyzed ligation vs. maleimide conjugation. The graphs show representative chromatograms obtained by SEC-ICP-MS analysis. Evaluation of the results was performed using the Qtegra^TM^ software. (**A**) [^175^Lu]Lu-DOTAGA-LPETGG-anti-PSMA-sdAb; (**B**) [^175^Lu]Lu-DOTAGA-cys-mal-anti-PSMA-sdAb-GGC.

**Figure 3 pharmaceutics-16-00299-f003:**
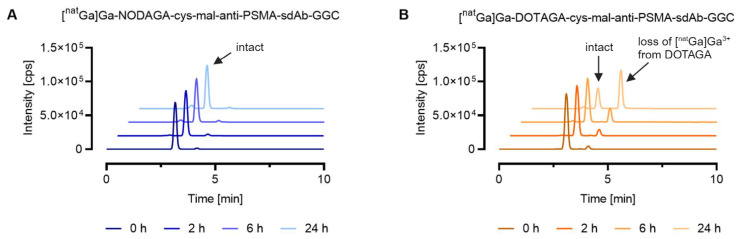
Comparison of the stability of ^nat^Ga-labeled anti-PSMA-sdAb conjugates equipped with two different metal–chelator complexes conjugated via maleimide conjugation. The graphs show representative chromatograms analyzed by SEC-ICP-MS. Evaluation of the results was performed using the Qtegra^TM^ software. (**A**) [^nat^Ga]Ga-NODAGA-cys-mal-anti-PSMA-sdAb-GGC; (**B**) [^nat^Ga]Ga-DOTAGA-cys-mal-anti-PSMA-sdAb-GGC.

**Figure 4 pharmaceutics-16-00299-f004:**
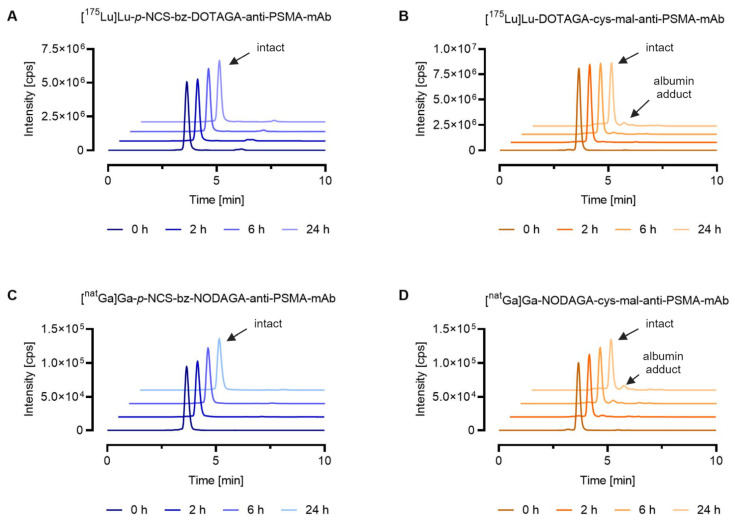
Stability of metal-labeled mAb conjugates in mouse serum at 37 °C up to 24 h. The graph shows representative chromatograms obtained by SEC-ICP-MS analysis. Evaluation of the results was conducted using the Qtegra^TM^ software. (**A**) [^175^Lu]Lu-*p*-NCS-bz-DOTAGA-anti-PSMA-mAb; (**B**) [^175^Lu]Lu-DOTAGA-cys-mal-anti-PSMA-mAb; (**C**) [^nat^Ga]Ga-*p*-NCS-bz-NODAGA-anti-PSMA-mAb; (**D**) [^nat^Ga]Ga-NODAGA-cys-mal-anti-PSMA-mAb.

**Table 1 pharmaceutics-16-00299-t001:** Summary of the typical operating conditions for ICP-MS.

ICP-MS	
Radiofrequency (RF) plasma power	1550 Watts
Plasma gas	Argon
Nebulizer gas flow	1.0 L/min
Nebulizer	PFA LC integrated capillary valve nebulizer (Elemental Scientific, Inc., Ohama, NE, USA)
Spray chamber	Cyclonic quartz spray chamber (Thermo Fisher Scientific, Switzerland)
Spray chamber temperature	2.7 °C
Sample/skimmer cone	Platinum (RPC), nickel (SEC)
Dwell time	0.3 s
Collision gas	Helium
Collision gas flow	6.45 mL/min
Additional gas	Oxygen, 5% (only for RPC)
Detection mode	SQ-He (lutetium-175), SQ-KED (gallium-71)

## Data Availability

The data presented in this study are available on request from the corresponding author.

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
