# Peer review of "Liquid Chromatography ICP-MS to Assess the Stability of 175Lu- and natGa-Based Tumor-Targeting Agents towards the Development of 177Lu- and 68Ga-Labeled Radiopharmaceuticals"

_pharmaceutics, 2024, doi:10.3390/pharmaceutics16030299_

Round 1

Reviewer 1 Report

Comments and Suggestions for Authors

see comments attached

Reviewer 2 Report

Comments and Suggestions for Authors

More validation data are required to validate the cold complex formation. Metalic impurities need to be quantified and presented in the extra information.

Also, in vitro studies are required to check the integrity of the cold complexes. It would be nice if some comparison will be made with the corresponding radiolabeled complexes.

Reviewer 3 Report

Comments and Suggestions for Authors

Overall, the manuscript is well-organized, with a clear progression from introduction to conclusion. The in vivo stability of radio drugs for radiotherapy is extremely important in the aspect of the long therapeutical window and good effects.

The abstract succinctly encapsulates the study's purpose and outcomes. It effectively communicates the motivation behind the research, which is to establish ICP-MS coupled with HPLC as a surrogate method for radio-HPLC in the preclinical characterization of radiopharmaceuticals. The abstract presents a compelling case for the use of non-radioactive analogs, exemplified by the stability assessment of 175Lu- and natGa-labeled prostate-specific membrane antigen (PSMA)-targeting agents.

The introduction provides a comprehensive contextualization of the role of radiometals in nuclear oncology and the necessity of chelators in developing targeted radiopharmaceuticals. The challenges associated with radioactive materials in ligand development are well-articulated, setting the stage for the study's relevance.

The rationale for the research is well-founded, emphasizing the need for alternative characterization techniques. The introduction skillfully lays out the groundwork for the study by linking the challenges in ligand development to the development of non-radioactive analogs for experimental convenience.

The literature review is thorough, covering various conjugation technologies and highlighting the existing gaps. The introduction of ICP-MS as a potential surrogate method is justified by its sensitivity and ability to solely detect the metal payload, mimicking the detection scenario of radiometric methods. The inclusion of alternative techniques for in vitro and in vivo studies strengthens the overall argument.

The methods section is detailed, providing a clear roadmap for the study's execution. The choice of model compounds, namely PSMA-11 and PSMA-617, enhances the study's credibility, given their commercial success and prior radiometric characterization. The inclusion of both small molecules and biologics broadens the study's scope, and the rationale behind the selection of DOTAGA and NODAGA chelators is well-explained. The methods section adequately addresses the conjugation techniques used for sdAbs and mAbs, including the stability considerations for thiosuccinimide linkage.

The results section effectively presents data on the stability of 175Lu- and natGa-labeled agents, showcasing the applicability of HPLC-ICP-MS. The inclusion of figures and tables aids in the interpretation of the data, enhancing clarity for readers.

The results are interpreted coherently, drawing meaningful comparisons with radiometrically characterized compounds. The study convincingly demonstrates the reliability of HPLC-ICP-MS for assessing in vitro stability. The data on sdAbs and mAbs contribute to the study's completeness, reflecting the stability variations seen with different conjugation chemistries.

The discussion is robust, contextualizing the findings within the broader landscape of radiopharmaceutical development. It effectively bridges the gap between the study's outcomes and their potential implications for ligand development.

The discussion appropriately addresses the study's limitations, such as the focus on specific chelators, contributing to a nuanced interpretation of the findings. The suggestions for future research, including exploration beyond the chosen chelators, add depth to the discussion and acknowledge the study's contribution to ongoing research.

The conclusions succinctly summarize the study's primary findings, emphasizing the successful substitution of certain assays in a conventional industrial or academic setting using RPC- and SEC-ICP-MS.

The conclusions underscore the practical implications of the study, highlighting the potential benefits of implementing RPC- and SEC-ICP-MS for expediting ligand pre-screening and facilitating the selection of promising candidates for further evaluation involving radioactivity.

Overall Evaluation:

The manuscript is well-structured and thoughtfully presented. The study addresses a significant gap in the field, offering a practical alternative for preclinical characterization. The use of well-established model compounds and chelators enhances the study's credibility, and the inclusion of biologics broadens its applicability. The manuscript is suitable for publication, and minor revisions should focus on enhancing clarity in certain sections, particularly in the introduction, to make the content more accessible to a broader readership.

If possible, studies with live mice (IV injection and following serum analysis) would further strengthen the method and manuscript.

Comments on the Quality of English Language

I believe some minor revision is still needed regarding the English language.

For example,

In the title, I would suggest the authors at least switch the two words, labeled and based. And if possible may try to make the title and abstract more concise.

line 17, should be: a surrogate

line 35, should be:  ensuring

line 52, should be: the literature

line 68, should be: the selected

line 83, however should be but, or add a comma right after however

line 425, should be: radio-HPLC

The typos and errors are more than what I pointed out above, please double-check the manuscript before resubmission.
